# EDM: Equirectangular Projection-Oriented Dense Kernelized Feature Matching

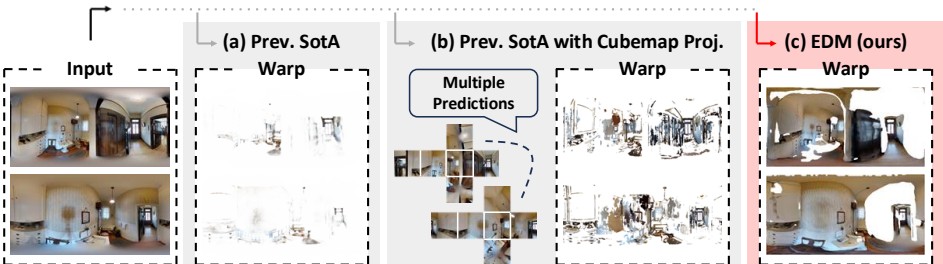

Figure 1: (a) Previous state-of-the-art (Edstedt et al., 2023a) struggles to achieve accurate dense matching in equirectangular projection (ERP) images due to inherent distortions. (b) The ERP image can be transformed into a cubemap image, which consists of six perspective images. However, this approach demands multiple independent iterations of inference for each pair of perspective images, increasing computational complexity and losing the global information in the ERP image. (c) Our proposed method, EDM, leverages the spherical camera model, rendering it robust against distortions. **Warp** refers to results obtained by multiplying the warped image with the predicted certainty map, demonstrating that our method yields more accurate dense matches.

## ABSTRACT

We introduce the first learning-based dense matching algorithm, termed Equirectangular Projection-Oriented Dense Kernelized Feature Matching (EDM), specifically designed for omnidirectional images. Equirectangular projection (ERP) images, with their large fields of view, are particularly suited for dense matching techniques that aim to establish comprehensive correspondences across images. However, ERP images are subject to significant distortions, which we address by leveraging the spherical camera model and geodesic flow refinement in the dense matching method. To further mitigate these distortions, we propose spherical positional embeddings based on 3D Cartesian coordinates of the feature grid. Additionally, our method incorporates bidirectional transformations between spherical and Cartesian coordinate systems during refinement, utilizing a unit sphere to improve matching performance. We demonstrate that our proposed method achieves notable performance enhancements, with improvements of +26.72 and +42.62 in AUC@5° on the Matterport3D and Stanford2D3D datasets, respectively.

## 1 INTRODUCTION

Omnidirectional images, also known as 360° images, provide significant advantages owing to their expansive fields of view, offering more contextual information and versatility (Xu et al., 2020; Zhang et al., 2023a; Matzen et al., 2017; da Silveira et al., 2022; Guerrero-Viu et al., 2020). These spherical images enable a comprehensive representation of environments, facilitating a deeper understanding of spatial information. Their utility extends to aiding robot navigation (Winters et al., 2000; Menegatti et al., 2004) and autonomous vehicle driving (Pandey et al., 2011) by minimizing blind spots. 360° images also can be utilized in a diverse range of applications, from creating immersive AR/VR experiences to practical uses in interior design (Amalia & Fitriyansah, 2023), tourism (Saurer et al., 2010), and real estate photography (Chang et al., 2017). Integrating omnidirectional images into virtual house tours allows customers to experience an immersive view, enabling them to fully engage themselves in the service. Moreover, the adoption of omnidirectional images contributes to more efficient data collection. By replacing the need for multiple perspective images, omnidirectional images can reduce both the cost and time associated with data scanning. The large

field of view provided by 360° images has also demonstrated superiority over narrower views in 3D motion estimation (Nelson & Aloimonos, 1988; Lee et al., 2000; Fermüller & Aloimonos, 2001).

Feature matching plays a critical role in numerous 3D computer vision tasks, including mapping and localization. Traditionally, Structure from Motion (SfM) (Schonberger & Frahm, 2016) leverages feature matching to estimate relative poses. Recent advancements have introduced semi-dense or dense approaches for feature matching such as LoFTR (Sun et al., 2021b) and DKM (Edstedt et al., 2023a), which demonstrate superior performance in repetitive or textureless environments compared to keypoint-based methods (Lowe, 2004; Rublee et al., 2011; DeTone et al., 2018; Sarlin et al., 2020; Li et al., 2022a). These methods have been mainly developed for perspective 2D images and videos, but encounter challenges when applied to omnidirectional images. For example, to adapt matching methods for spherical images, two prevalent approaches for sphere-to-plane projections are the equirectangular projection (ERP) and the cubemap projection (Xu et al., 2020). ERP images exhibit significant distortions, particularly near the pole regions, which hinder the effective application of perspective methods. On the other hand, the cubemap format, consisting of six perspective images, can be processed independently without such distortions. However, this approach involves the costly computation of multiple inferences for each pair of perspective images, resulting in the loss of global information from a single spherical image and diminishing feature matching capabilities due to the reduced field of view in each perspective image. These challenges are shown in Fig. 1 (a) and (b).

**Main Results**    In this paper, we propose EDM, a distortion-aware dense feature matching method for omnidirectional images, addressing challenges that existing detector-free approaches (Sun et al., 2021b; Edstedt et al., 2023a;b) struggle to overcome. To the best of our knowledge, EDM is the first learning-based method designed for dense matching and relative pose estimation between two omnidirectional images. As seen in Fig. 1, our method defines feature matching in 3D coordinates, specifically addressing the challenges posed by distortions of ERP images. We accomplish this based on the integration of two novel steps: a Spherical Spatial Alignment Module (SSAM) and specific enhancements in Geodesic Flow Refinement. The SSAM leverages spherical positional embeddings for ERP images and incorporates a decoder to generate the global matches. Furthermore, the Geodesic Flow Refinement step employs coordinate transformation to refine the residuals of correspondences. Compared to both recent sparse and dense feature matching methods (Zhao et al., 2015; Gava et al., 2023; Edstedt et al., 2023a;b), our approach results in significant performance improvement of +26.72 and +42.62 AUC@5° in relative pose estimation for spherical images on the Matterport3D (Chang et al., 2017) and Stanford2D3D (Armeni et al., 2017) datasets. Additionally, we evaluate our method qualitatively on the EgoNeRF (Choi et al., 2023) and OmniPhotos (Bertel et al., 2020) datasets, demonstrating robust performance across diverse environments. The main contributions of this paper are summarized as follows:

- We introduce a novel approach for estimating dense matching across ERP images using geodesic flow on a unit sphere.

- We propose a Spherical Spatial Alignment Module that utilizes Gaussian Process regression and spherical positional embeddings to establish 3D correspondences between omnidirectional images. In addition, we use Geodesic Flow Refinement by enabling conversions between coordinates to refine the displacement on the surface of the sphere.

- With azimuth rotation for data augmentation, we achieve state-of-the-art performance in dense matching and relative pose estimation between two omnidirectional images.

## 2    RELATED WORK

**Omnidirectional Images**    The popularity of consumer-level 360° cameras has led to increased interest in spherical images, which offer comprehensive coverage of the field of view from a single vantage point. These images are often represented using equirectangular projection (ERP) (Xu et al., 2020), facilitating their utilization in various computer vision tasks. Recent advancements in computer vision have leveraged ERP images for diverse tasks such as object detection (Coors et al., 2018; Su & Grauman, 2017), semantic segmentation (Jiang et al., 2019; Zhang et al., 2019), depth estimation (Jiang et al., 2021; Wang et al., 2020; Shen et al., 2022; Li et al., 2022b; Rey-Area et al., 2022; Li et al., 2021; Yun et al., 2022), omnidirectional Simultaneous Localization and Mapping

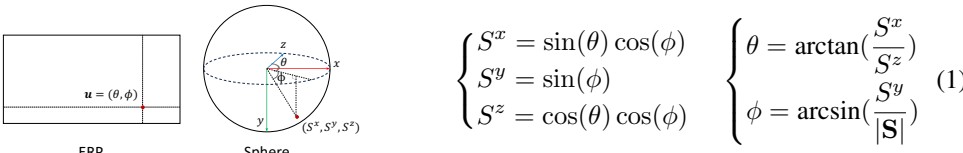

$$\begin{cases} S^x = \sin(\theta)\cos(\phi) \\ S^y = \sin(\phi) \\ S^z = \cos(\theta)\cos(\phi) \end{cases} \quad \begin{cases} \theta = \arctan(\dfrac{S^x}{S^z}) \\ \phi = \arcsin(\dfrac{S^y}{|\mathbf{S}|}) \end{cases} \quad (1)$$

Figure 2: Coordinate system.

(Won et al., 2020), scene understanding (Sun et al., 2021a), and neural rendering (Choi et al., 2024; Kim et al., 2024; Ma et al., 2024; Li et al., 2024).

Despite the utility of ERP images, their unique geometry presents several challenges in visual representation. As ERP images are obtained through projecting a sphere onto a plane, a single spherical image can be expressed by multiple distinct ERP images. Additionally, ensuring perfect alignment of their left and right extremities is essential. While some research methods have introduced rotation-equivariant convolutions (Cohen et al., 2018; Esteves et al., 2018) to address these issues, their implementation often demands increased computational resources. To mitigate this constraint, we propose an azimuth rotation approach for data augmentation, under the assumption that maintaining the downward orientation of scanned omnidirectional images parallel to gravity offers benefits (Bergmann et al., 2021).

**Feature Matching** Local feature matching has relied on detector-based methods, encompassing both traditional hand-crafted techniques (Lowe, 2004; Rublee et al., 2011) and learning-based approaches (DeTone et al., 2018; Revaud et al., 2019; Li et al., 2022a; Liu et al., 2019; Tyszkiewicz et al., 2020). These methods typically involve detecting keypoints, computing descriptor distances between paired keypoints, and performing matching via mutual nearest neighbor search. SuperGlue (Sarlin et al., 2020) introduces a learning-based paradigm, optimizing visual descriptors using an attentional graph neural network and an optimal matching layer. However, detector-based methods face limitations in terms of accurately detecting keypoints, particularly in repetitive or indiscriminative regions. In contrast, detector-free or dense methods (Sun et al., 2021b; Melekhov et al., 2019; Truong et al., 2020; 2021; Edstedt et al., 2023a;b) offer a solution to the keypoint detection issue, providing dense feature matches at the pixel level.

While the aforementioned methods are tailored for perspective images, they often fail to address the unique challenges of spherical cameras. SPHORB (Zhao et al., 2015), an extension of ORB (Rublee et al., 2011), mitigates distortion in ERP images using a geodesic grid and local planar approximation (Eder et al., 2020). Similarly, learning-based matching methods such as SphereGlue (Gava et al., 2023; 2024) and PanoPoint (Zhang et al., 2023b) adapt keypoint matching techniques for spherical imagery. CoVisPose (Hutchcroft et al., 2022; Nejatishahidin et al., 2023) explores layout features for estimating camera poses over large baselines yet remains constrained by detected feature information. Therefore, we propose a novel dense matching method that extracts all matches without keypoint detection in spherical images.

## 3 PRELIMINARIES

### 3.1 SPHERICAL AND CARTESIAN COORDINATE

Although ERP images are displayed in 2D space, they actually represent a collection of flattened rays normalized to a unit scale within a spherical camera model. Thus, we can express the coordinate conversion equation $\mathbf{u} = \boldsymbol{\pi}(\mathbf{S})$ between the spherical coordinates $\mathbf{u} = (\theta, \phi)$ and the 3D Cartesian coordinates $\mathbf{S} = (S^x, S^y, S^z)$ as shown in Fig. 2. Each value of $\theta \in [-\pi, \pi]$ and $\phi \in [-\frac{\pi}{2}, \frac{\pi}{2}]$ indicates the longitude and latitude. We utilize this coordinate transformation $\boldsymbol{\pi}(\cdot)$ in Section 4.1 and Section 4.2 to handle the spherical camera model effectively.

### 3.2 DENSE KERNELIZED FEATURE MATCHING

Dense matching is the task of finding dense correspondence and estimating 3D geometry from two images $(I_\mathcal{A}, I_\mathcal{B})$. Recently, DKM (Edstedt et al., 2023a) introduced a kernelized global matcher and warp refinement, formulating this problem as finding a mapping $f \to \mathbf{u}$ where $\mathbf{u}$ are 2D spatial

coordinates. First, DKM extracts multi-scale features using a ResNet50 encoder (He et al., 2016),

$$\{f_{\mathcal{A}}^l\}_{l=1}^L = \text{Encoder}(I_{\mathcal{A}}), \quad \{f_{\mathcal{B}}^l\}_{l=1}^L = \text{Encoder}(I_{\mathcal{B}}), \tag{2}$$

where the strides are defined as elements of the set $l \in \{2^0, ..., 2^{L-1}\}$. Coarse features are associated with stride $\{32, 16\}$, and fine features correspond to $\{8, 4, 2, 1\}$.

At the coarse level, it consists of a kernelized regression to estimate the posterior mean $\mu_{\mathcal{A}|\mathcal{B}}$ using a Gaussian Process (GP) formulation. GP regression generates a probabilistic distribution using the feature information conditioned on frame $\mathcal{B}$ to estimate coarse global matches. The normalized 2D feature grid $f_{\mathcal{B}}^{\text{grid}} \in \mathbb{R}^{h \times w \times 2}$, where $h$ and $w$ denote the resolution of the feature grid, is embedded into $\chi_{\mathcal{B}}$ with an additional cosine embedding (Snippe & Koenderink, 1992) to induce multimodality in GP. The embedded coordinates are processed by an exponential cosine similarity kernel $K$ to calculate $\mu_{\mathcal{A}|\mathcal{B}}$,

$$\mu_{\mathcal{A}|\mathcal{B}} = K_{\mathcal{A}\mathcal{B}}(K_{\mathcal{B}\mathcal{B}} + \sigma_n^2 I)^{-1} \chi_{\mathcal{B}}^{\text{coarse}}, \tag{3}$$

$$\begin{cases} K_{mn} = \exp\left(\tau\left(\dfrac{f_m \cdot f_n}{\sqrt{(f_m \cdot f_m)(f_n \cdot f_n) + \varepsilon}} - 1\right)\right), \\ \chi_{\mathcal{B}}^{\text{coarse}} = \cos(W f_{\mathcal{B}}^{\text{grid}} + b), \end{cases} \tag{4}$$

where $\tau = 5$, $\epsilon = 10^{-6}$, and the standard deviation of the measurement noise $\sigma_n = 0.1$ in the experiments. $W$ and $b$ are the weights and biases of a $1 \times 1$ convolution layer. Then, CNN embedding decoder (Yu et al., 2018) yields the initial global matches $\hat{\mathbf{u}}_{\mathcal{A}\to\mathcal{B}}^{\text{coarse}}$ and confidence of matches $\hat{c}_{\mathcal{A}\to\mathcal{B}}^{\text{coarse}}$ from the concatenation of the reshaped estimated posterior mean $\mu_{\mathcal{A}|\mathcal{B}}^{\text{grid}}$ and the coarse features,

$$(\hat{\mathbf{u}}_{\mathcal{A}\to\mathcal{B}}^{\text{coarse}}, \hat{c}_{\mathcal{A}\to\mathcal{B}}^{\text{coarse}}) = \text{Decoder}(\mu_{\mathcal{A}|\mathcal{B}}^{\text{grid}} \oplus f_{\mathcal{A}}^{\text{coarse}}). \tag{5}$$

At the fine level, the warp refiners estimate the residual displacement using the previous matches and feature information. The process is described as follows,

$$\left(\triangle\hat{\mathbf{u}}_{\mathcal{A}\to\mathcal{B}}^{l+1}, \triangle\hat{c}_{\mathcal{A}\to\mathcal{B}}^{l+1}\right) = \text{Refiner}^{l+1}\left(f_{\mathcal{A}}^{l+1} \oplus f_{\mathcal{B}\to\mathcal{A}}^{l+1} \oplus Corr_{\Omega_k}^{l+1} \oplus \hat{\mathbf{u}}_{\mathcal{A}\to\mathcal{B}}^{l+1} - \mathbf{u}_{\mathcal{A}}^{l+1}\right), \tag{6}$$

$$\begin{cases} f_{\mathcal{B}\to\mathcal{A}}^{l+1} = f_{\mathcal{B}}\langle\hat{\mathbf{u}}_{\mathcal{A}\to\mathcal{B}}^{l+1}\rangle, \quad f_{\mathcal{B}\to\mathcal{A},\,\Omega_k}^{l+1} = f_{\mathcal{B}}\langle\Omega_k, (\hat{\mathbf{u}}_{\mathcal{A}\to\mathcal{B}}^{l+1})\rangle, \\ Corr_{\Omega_k}^{l+1} = \sum_{\text{channel}} f_{\mathcal{A}}^{l+1}\, f_{\mathcal{B}\to\mathcal{A},\,\Omega_k}^{l+1}, \end{cases} \tag{7}$$

where $\Omega_k(\mathbf{u}) = \mathbf{u} + \mathbf{p}$ ($\|\mathbf{p}\|_\infty \leq k$) is the patch sized $k$, $\langle\cdot\rangle$ means the bilinear interpolation function, $Corr_{\Omega_k}^{l+1}$ represents local correlation between the features, and $\mathbf{u}_{\mathcal{A}}^{l+1}$ indicates the grid in $f_{\mathcal{A}}^{l+1}$. Finally, it recursively updates the matching points and confidence by adding the residuals to the previous information and upsampling until reaching the same resolution as the input images,

$$\hat{\mathbf{u}}_{\mathcal{A}\to\mathcal{B}}^l = \hat{\mathbf{u}}_{\mathcal{A}\to\mathcal{B}}^{l+1} + \triangle\hat{\mathbf{u}}_{\mathcal{A}\to\mathcal{B}}^{l+1}, \quad \hat{c}_{\mathcal{A}\to\mathcal{B}}^l = \hat{c}_{\mathcal{A}\to\mathcal{B}}^{l+1} + \triangle\hat{c}_{\mathcal{A}\to\mathcal{B}}^{l+1}. \tag{8}$$

## 4 OUR PROPOSED METHOD

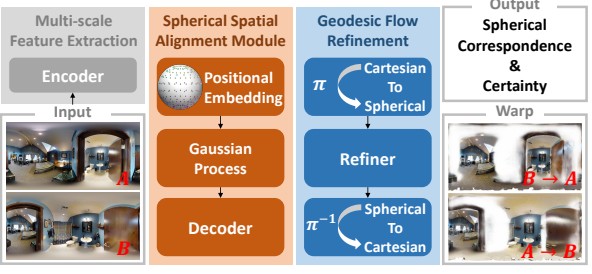

Figure 3: Overview of our approach. It consists of three steps: Multi-scale Feature Extraction, Spherical Spatial Alignment Module (Sec.4.1), and Geodesic Flow Refinement (Sec.4.2).

The overall process is illustrated in Fig. 3. Following the approach outlined in Section 3.2, we first utilize ERP images $I_{\mathcal{A}}$ and $I_{\mathcal{B}}$ as input and extract multi-scale features $f_{\mathcal{A}}$ and $f_{\mathcal{B}}$. Different from (Edstedt et al., 2023a), we reformulate the problem as finding a mapping $f \to \mathbf{S}$ using 3D Cartesian coordinates. We introduce the Spherical Spatial Alignment Module, a global matcher utilizing a spherical camera system to compensate for distortions caused by sphere-to-plane projection in ERP images. We then formalize the geodesic flow on a unit sphere and establish projections between equirectangular and spherical spaces to refine matches. In addition, to enhance the robust accuracy of our method, we leverage randomized azimuth rotation during the training process.

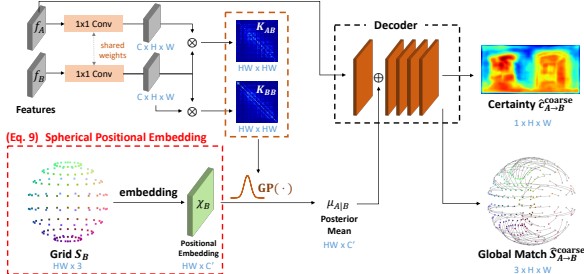

Figure 4: Our Spherical Spatial Alignment Module. We present Spherical Positional Embedding (red dotted box). The embedding decoder generates the global matches $\hat{\mathbf{S}}_{\mathcal{A}\to\mathcal{B}}^{\text{coarse}}$. Here, the gray curved lines represent the geodesic flow between $\mathbf{S}_{\mathcal{A}}$ and $\mathbf{S}_{\mathcal{B}}$. $\oplus$ denotes concatenation, $\otimes$ means reshape and matrix multiplication. We provide the matrix dimensions of intermediate features for reference.

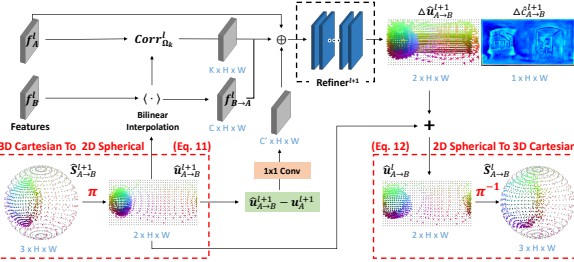

Figure 5: Our proposed Geodesic Flow Refinement. Refining the displacement along curved lines on the spherical surface presents significant challenges. To address this, we project the displacement into the ERP space for refinement (Cartesian to spherical) and subsequently unproject it back onto the spherical surface for further refinement (spherical to Cartesian).

## 4.1 SPHERICAL SPATIAL ALIGNMENT MODULE

Our Spherical Spatial Alignment Module (SSAM) conducts global matching at a coarse level through Gaussian Process (GP) regression, depicted in Fig. 4. GP predicts the posterior mean $\mu_{\mathcal{A}|\mathcal{B}}$ from the embeddings as in Eq. 3. Due to the pronounced distortions in the polar regions of ERP images, spherical positional embedding/encoding is frequently employed to mitigate this challenge (Chen et al., 2022; Li et al., 2023a;b). Here, we explicitly apply positional embeddings with 3D Cartesian coordinates, derived from the 2D spherical feature grid and the inverse transformation function $\boldsymbol{\pi}^{-1}(\cdot)$,

$$\chi_{\mathcal{B}}^{\text{coarse}} = \cos(W\boldsymbol{\pi}^{-1}(f_{\mathcal{B}}^{\text{grid}}) + b). \tag{9}$$

Our proposed positional embedding facilitates the utilization of embedded coordinates $\chi_{\mathcal{B}}^{\text{coarse}}$ to promote distortion awareness within the ERP images. Additionally, this embedding ensures structural consistency along the boundaries of ERP images by leveraging relative spatial information within the 3D Cartesian grid. The outputs of the subsequent embedding decoder provide the initial global matches $\hat{\mathbf{S}}_{\mathcal{A}\to\mathcal{B}}^{\text{coarse}}$ on the unit sphere and the ERP certainty map $\hat{c}_{\mathcal{A}\to\mathcal{B}}^{\text{coarse}}$,

$$\left(\hat{\mathbf{S}}_{\mathcal{A}\to\mathcal{B}}^{\text{coarse}}, \hat{c}_{\mathcal{A}\to\mathcal{B}}^{\text{coarse}}\right) = \text{Decoder}(\mu_{\mathcal{A}|\mathcal{B}} \oplus f_{\mathcal{A}}^{\text{coarse}}). \tag{10}$$

## 4.2 GEODESIC FLOW REFINEMENT

In our SSAM approach, as the geodesic flow must reside on the unit sphere, directly defining warp refinement on the surface of the sphere makes it impossible to update the residuals linearly. Thus, we circumvent this problem by enabling a conversion between the 3D Cartesian coordinates and the 2D equirectangular space, as illustrated in Fig. 5,

$$\hat{\mathbf{u}}_{\mathcal{A}\to\mathcal{B}}^{l+1} = \boldsymbol{\pi}(\hat{\mathbf{S}}_{\mathcal{A}\to\mathcal{B}}^{l+1}). \tag{11}$$

After following all the processes outlined in Eq. 6 for refinement, we update the residuals as described in Eq. 8. As this refinement stage iterates repeatedly, the predicted $\hat{\mathbf{u}}_{\mathcal{A}\to\mathcal{B}}^{l}$ is back-projected into 3D Cartesian coordinates,

$$\hat{\mathbf{S}}_{\mathcal{A}\to\mathcal{B}}^{l} = \boldsymbol{\pi}^{-1}(\hat{\mathbf{u}}_{\mathcal{A}\to\mathcal{B}}^{l}). \tag{12}$$

Figure 6: Maintaining consistent geometry, ERP can produce multiple visual representations based on $\theta^{\text{aug}}$.

$$\begin{cases} I_{\mathcal{A}} & \leftarrow & I_{\mathcal{A}} \left\langle \boldsymbol{\pi} \left( T_{\mathcal{A}}^{\text{aug}} \boldsymbol{\pi}^{-1}(I_{\mathcal{A}}^{\text{grid}}) \right) \right\rangle \\ D_{\mathcal{A}} & \leftarrow & D_{\mathcal{A}} \left\langle \boldsymbol{\pi} \left( T_{\mathcal{A}}^{\text{aug}} \boldsymbol{\pi}^{-1}(D_{\mathcal{A}}^{\text{grid}}) \right) \right\rangle \\ T_{\mathcal{A}} & \leftarrow & T_{\mathcal{A}} \, T_{\mathcal{A}}^{\text{aug}} \end{cases} \quad (13)$$

## 4.3 AUGMENTATION

A single omnidirectional image can be transformed into multiple distinct ERP images, as shown in Fig. 6. This transformation is feasible by capturing the full spectrum of rays and ensuring a seamless representation in the spherical input image, which facilitates the generation of diverse ERP images while maintaining consistent geometric properties in the world space. Consequently, we define a horizontal rotation matrix $T_{\mathcal{A}}^{\text{aug}}$ with a randomly selected azimuth angle $\theta_{\mathcal{A}}^{\text{aug}} \in [0, 2\pi]$ during training. Based on $T_{\mathcal{A}}^{\text{aug}}$, we rotate and redefine the ERP image $I_{\mathcal{A}}$, the depth map $D_{\mathcal{A}}$, and the pose $T_{\mathcal{A}}$ as specified in Eq. 13. Notably, this transformation adjusts $T_{\mathcal{A}}$ and $D_{\mathcal{A}}$ together, ensuring consistent geometry in the world space. The same process is applied to the counterpart frame $\mathcal{B}$.

## 4.4 LOSS

Utilizing dense ground truth depth maps and aligned camera poses, we can derive ERP depth $D_{\mathcal{A} \to \mathcal{B}}$ and matches $\mathbf{S}_{\mathcal{A} \to \mathcal{B}}$ during the warping process from frame $\mathcal{A}$ to $\mathcal{B}$ within the spherical coordinate system. We adopt the certainty estimation method proposed by Edstedt et al. (2023a), which involves finding consistent matches using relative depth consistency between frames $\mathcal{A}$ and $\mathcal{B}$,

$$c_{\mathcal{A} \to \mathcal{B}} = \left| \frac{D_{\mathcal{A} \to \mathcal{B}} - D_{\mathcal{B}}}{D_{\mathcal{B}}} \right| < \alpha, \quad (14)$$

where $\alpha$ is 0.05. The binary mask $c_{\mathcal{A} \to \mathcal{B}}$ represents the ground truth certainty map. Diverging from the approach outlined in Edstedt et al. (2023a), our method constrains the predicted matches $\hat{\mathbf{S}}_{\mathcal{A} \to \mathcal{B}}^l$, composed of 3D Cartesian coordinates, to reside on the surface of the unit sphere. This implies that the predicted matches can be interpreted as the ray directions of the spherical camera. Instead of defining the loss function based on the Euclidean distance between the predicted matches $\hat{\mathbf{S}}_{\mathcal{A} \to \mathcal{B}}^l$ and the ground truth matches $\mathbf{S}_{\mathcal{A} \to \mathcal{B}}^l$, we use the angular difference between the ray directions. Consequently, this approach ensures that $\hat{\mathbf{S}}_{\mathcal{A} \to \mathcal{B}}^l$ is optimized along the surface of the unit sphere. We define our regression loss $L_{\text{r}}^l$ using cosine similarity to measure the angular difference. For the certainty loss $L_{\text{c}}^l$, we employ the binary cross-entropy function, as utilized in Edstedt et al. (2023a),

$$\mathcal{L}_{\text{r}}^l = \sum_{\text{grid}} c_{\mathcal{A} \to \mathcal{B}}^l \odot (1 - \frac{\|\mathbf{S}_{\mathcal{A} \to \mathcal{B}}^l \cdot \hat{\mathbf{S}}_{\mathcal{A} \to \mathcal{B}}^l\|}{\|\mathbf{S}_{\mathcal{A} \to \mathcal{B}}^l\| \|\hat{\mathbf{S}}_{\mathcal{A} \to \mathcal{B}}^l\|}), \quad (15)$$

$$\mathcal{L}_{\text{c}}^l = \sum_{\text{grid}} c_{\mathcal{A} \to \mathcal{B}}^l \log \hat{c}_{\mathcal{A} \to \mathcal{B}}^l + (1 - c_{\mathcal{A} \to \mathcal{B}}^l) \log(1 - \hat{c}_{\mathcal{A} \to \mathcal{B}}^l). \quad (16)$$

The total loss function comprises a weighted sum of the regression loss and the certainty loss, as detailed in Zhou et al. (2021); Melekhov et al. (2019); Tan et al. (2022); Edstedt et al. (2023a), with $\lambda$ set at 0.01,

$$L_{\text{total}} = \sum_{l=1}^{L} L_{\text{r}}^l + \lambda L_{\text{c}}^l. \quad (17)$$

## 5 EXPERIMENTS

### 5.1 EXPERIMENTS SETTINGS

**Matterport3D Dataset** Training our method requires ERP input images, ground truth depth maps, and aligned poses. The Matterport3D dataset (Chang et al., 2017) encompasses 90 indoor scenes

represented by 10,800 panoramas reconstructed as textured meshes. However, the dataset lacks pose and depth information for *skybox* images, which are essential for creating ERP images. Previous works have addressed this limitation by rendering both images and depth maps from the textured mesh (Zioulis et al., 2018) or by employing 360° SfM to estimate poses (Rey-Area et al., 2022). In our approach, we generate the poses for *skybox* images directly from the originally proposed camera poses in Matterport3D. Through experimentation, we found that treating the 12th camera pose, out of the 18 viewpoints (comprising 6 rotations and 3 tilt angles) in each panorama, identically to the second skybox image did not result in any issues. We define the remaining poses for the *skybox* images by rotating 90° in each direction from the second pose. We adhere to the official benchmark split, utilizing 61 scenes for training, 11 for validation, and 18 for testing. For two-view pose estimation, it is necessary to create pairs of overlapped images. We achieve this by transforming ERP depth maps between frames within the spherical coordinate system. Pixels where the depth difference is below a specified threshold, e.g. 0.1, are classified as inliers. Subsequently, we compare the ratio of these inliers to the total number of pixels. We organize both the training and testing datasets based on the overlap ratio of image pairs and the benchmark split. Specifically, images with the overlap ratio exceeding 30% are distributed into respective training and testing splits. As a result, the training set contains 44,700 pairs, while the test set comprises 4,575 pairs. We resize the resolution of ERP images and depth maps to $640 \times 320$.

**Stanford2D3D Dataset**    Stanford2D3D (Armeni et al., 2017) consists of data scanned from six large-scale indoor spaces collected from three distinct buildings. This dataset contains a relatively small number of 1,413 panorama images and, therefore, is utilized exclusively for testing purposes. We assess the overlap ratio between frames and include them in the test split if their ratio exceeds 50%. A total of 3,460 pairs are incorporated into the test set. During testing, we resize the resolution to $640 \times 320$.

**EgoNeRF and OmniPhotos Dataset**    EgoNeRF (Choi et al., 2023) introduces 11 synthetic scenes created with Blender (Community, 2018) and 11 real scenes captured with a RICOH THETA V camera. OmniPhotos (Bertel et al., 2020) provides a dataset captured with an Insta360 ONE X camera. Both datasets contain egocentric scenes captured with a casually rotating camera stick. Consequently, their rotation axes, pole regions, or camera height change, resulting in different distortions compared to Matterport3D or Stanford2D3D. We present additional qualitative results from these datasets to validate our method.

**Implementation Details**    We employ the AdamW (Loshchilov & Hutter, 2017) optimizer with a weight-decay factor of $10^{-2}$, a learning rate of $5 \cdot 10^{-6}$ for multiscale feature extractor, and $10^{-4}$ for the SSAM and the Geodesic Flow Refiner. EDM is trained for 300,000 steps with a batch size of 4 in a single RTX 3090 GPU, which takes approximately two days to complete. During evaluation, the balanced sampling approach using kernel density estimation (Edstedt et al., 2023a) tends to establish correspondences primarily in concentrated areas with high probability distributions, making it unsuitable for omnidirectional images. Thus, we randomly sample up to 5,000 matches after certainty filtering with a threshold of 0.8 to ensure correspondences cover the entire area.

## 5.2 EXPERIMENTAL RESULTS

We compare our proposed method EDM with four different methods: 1) SPHORB (Zhao et al., 2015) is a hand-crafted keypoint-based feature matching algorithm. 2) SphereGlue (Gava et al., 2023) is a learning-based keypoint matching method. Both SPHORB (Zhao et al., 2015) and SphereGlue (Gava et al., 2023) are specifically designed for spherical images. 3) DKM (Edstedt et al., 2023a) and 4) RoMa (Edstedt et al., 2023b) are state-of-the-art dense matching algorithms for perspective images. To estimate the essential matrix and the relative pose for spherical cameras, Solarte et al. (2021) proposed a normalization strategy and non-linear optimization within the classic 8-point algorithm. We adopt this for two-view pose estimation in all quantitative comparisons.

Table 1 shows the quantitative results of the pose estimation in Matterport3D. Despite SPHORB and SphereGlue being designed for the ERP images, the presence of textureless or repetitive regions, which are common in indoor environments of Matterport3D, leads to performance degradation in the keypoint-based methods. SPHORB fails to estimate the essential matrix correctly due to the limited number of matching points. EDM demonstrates significantly higher performance than all the other methods.

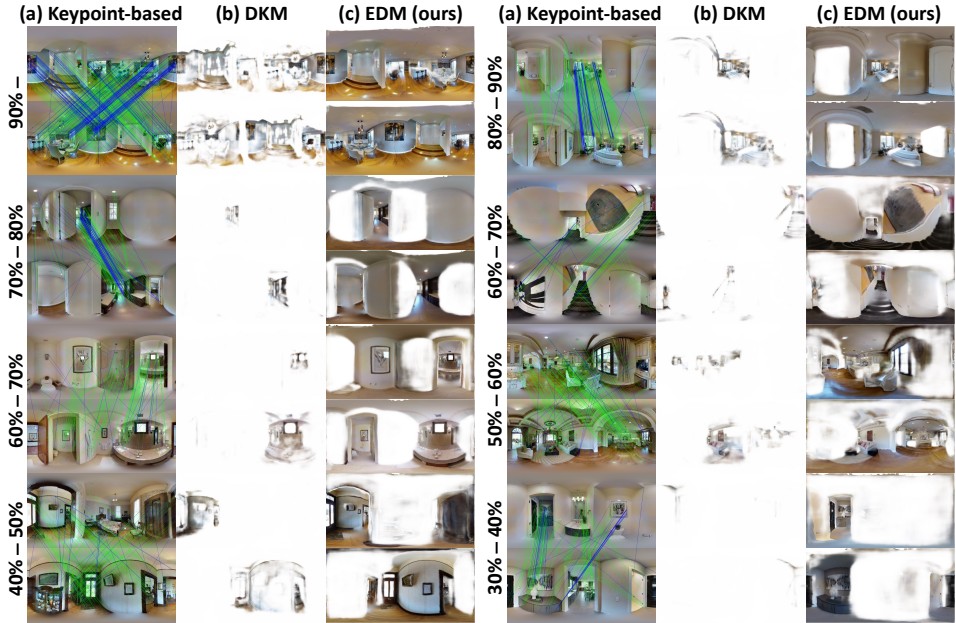

Figure 7: Qualitative results on Matterport3D. (a) The blue lines represent the results of matching points from SPHORB; the green lines correspond to SphereGlue. Both (b) DKM and (c) EDM depict the outcomes of multiplying the warped image with the certainty map. EDM can estimate dense and accurate matches even in the presence of distortions and severe occlusions. The numbers beside the images represent the overlap ratio, reflecting the difficulty of matching. Smaller numbers indicate more challenging scenes.

Table 1: Quantitative comparison on Matterport3D. EDM improve AUC@5° by 26.72.

| Method | Image | Feature | AUC (%) ↑ @5° | @10° | @20° |
|--------|-------|---------|------|------|------|
| SPHORB (Zhao et al., 2015) | ERP | sparse | 0.38 | 1.41 | 3.99 |
| SphereGlue (Gava et al., 2023) | ERP | sparse | 11.29 | 19.95 | 31.10 |
| DKM (Edstedt et al., 2023a) | perspective | dense | 18.43 | 28.50 | 38.44 |
| RoMa (Edstedt et al., 2023b) | perspective | dense | 12.45 | 22.37 | 34.24 |
| **EDM (ours)** | ERP | dense | **45.15** | **60.99** | **73.60** |

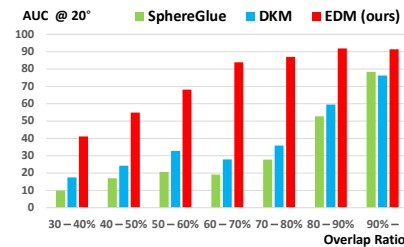

Figure 8: Performance relative to the overlap ratio.

Figure 7 illustrates the qualitative results in Matterport3D. The previous methods designed for perspective images, such as DKM and RoMa, exhibit good matching ability but encounter challenges when confronted with the distortions of ERP. While SphereGlue and SPHORB perform well in discriminative regions, their performance deteriorates as the overlap ratio decreases, resulting in numerous false positive matches. In contrast, EDM can estimate dense correspondences regardless of occlusion and textureless areas. Due to the similarity in results between DKM and RoMa, we have only included the former to maintain a concise visualization. Experimental results in Fig. 8 depict the relationship between image overlap ratio and AUC@20° performance. As expected, a decrease in the overlap ratio leads to severe performance degradation in the previous works. On the other hand, our proposed method demonstrates robustness in more challenging scenes, maintaining similar performance levels until the overlap decreases to 60%, compared to other methods.

For a fair comparison, we use another benchmark dataset, Stanford2D3D. We validate EDM using a model trained on Matterport3D without additional training on Stanford2D3D. In Table 2, EDM outperforms the previous works by a significant margin, especially in scenes with severe occlusion. The certainty map demonstrates EDM's robustness, particularly in handling occluded scenes. Additionally, although the panorama images in Stanford2D3D contain missing regions in the upper and lower parts of the sphere, the proposed spherical positional embedding enables the network to predict matching correspondences accurately, as shown in Fig. 9.

Table 2: Quantitative comparison on Stanford2D3D. EDM improve AUC@5° by 42.62.

| Method | Image | Feature | AUC (%) ↑ | | |
|---|---|---|---|---|---|
| | | | @5° | @10° | @20° |
| SPHORB (Zhao et al., 2015) | ERP | sparse | 0.14 | 1.01 | 4.08 |
| SphereGlue (Gava et al., 2023) | ERP | sparse | 11.25 | 22.41 | 36.57 |
| DKM (Edstedt et al., 2023a) | perspective | dense | 12.46 | 22.18 | 34.13 |
| RoMa (Edstedt et al., 2023b) | perspective | dense | 11.48 | 22.52 | 37.07 |
| **EDM (ours)** | ERP | dense | **55.08** | **71.65** | **82.72** |

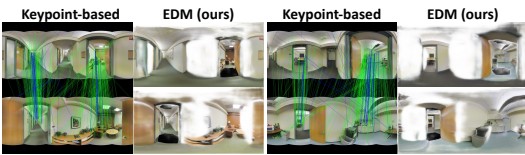

Figure 9: Qualitative results on Stanford2D3D. The blue and green lines correspond to SPHORB and SphereGlue.

### 5.3 ADDITIONAL QUALITATIVE RESULTS

To demonstrate the robust performance of our method across diverse environments, we qualitatively validate EDM using additional datasets such as EgoNeRF and OmniPhotos. As it is primarily trained on indoor environments (Chang et al., 2017) where the camera is oriented parallel to gravity, severely slanted image pairs of rotational scenes or outdoor environments may cause EDM to fail in accurately estimating correspondences. However, despite these differences in settings, EDM demonstrates the ability to conduct dense feature matching robustly, as shown in Fig. 10.

### 5.4 ABLATION STUDY

DKM's dependence on the pinhole camera model makes it inherently unsuitable for learning with ERP images. To ensure the fair comparison, we modified the warping process in the loss function of DKM to support spherical cameras, resulting in DKM*. As shown in Table 3, this demonstrates the structural effectiveness of our proposed bidirectional coordinate transformation. The proposed positional embeddings result in improvements based on the coordinate system of the spherical camera model. We observe that utilizing a 3D grid input of Cartesian coordinates yields better performance than 2D spherical ones. Additionally, in our method, positional embedding with a linear layer slightly outperforms spherical positional encoding with sinusoidal (Li et al., 2023b). Table 3 also confirms the advantage of our rotational augmentation. Through this augmentation technique, we can effectively address the challenge of a limited number of datasets for omnidirectional images in dense matching tasks.

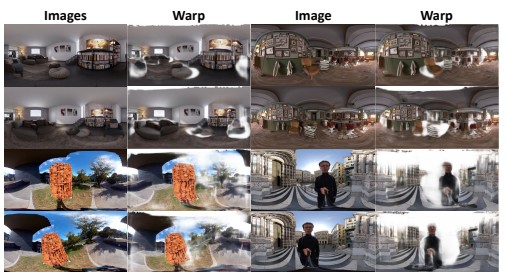

Figure 10: Qualitative results on EgoNeRF and OmniPhotos.

Table 3: Ablation study for the proposed method. DKM* indicates the DKM model trained on Matterport3D with a modified loss function for ERP images. Compared to DKM*, our method enhances performance through the proposed spherical positional embedding in SSAM, bidirectional transformation via Geodesic Flow Refinement, and rotational augmentation.

| Method | Positional Embedding | Bidirectional Transformation | Rotational Augmentation | AUC | | |
|---|---|---|---|---|---|---|
| | | | | @5° | @10° | @20° |
| DKM* | 2D linear | - | - | 19.83 | 33.06 | 46.24 |
| Ours | 2D linear | ✓ | - | 29.67 | 45.90 | 60.82 |
| Ours | 2D linear | ✓ | ✓ | 35.03 | 51.14 | 65.07 |
| Ours | 3D linear | ✓ | - | 34.64 | 50.82 | 65.16 |
| **Ours** | **3D linear** | ✓ | ✓ | **45.15** | **60.99** | **73.60** |
| Ours | 3D sinusoidal | ✓ | ✓ | 42.39 | 58.27 | 70.98 |

## 6 CONCLUSION, LIMITATIONS, AND FUTURE WORK

In this paper, we present, for the first time, a novel dense feature matching method tailored for omnidirectional images. Leveraging the foundational principles of DKM, we integrate the inherent characteristics of the spherical camera model into our dense matching process using geodesic flow fields. This integration instills distortion awareness within the network, thereby enhancing its performance specifically for ERP images. However, it is important to note that our method is predominantly trained on indoor datasets where the camera is vertically oriented, rendering it somewhat vulnerable to extreme rotations or outdoor environments. To address this limitation, future endeavors will focus on diversifying the training data and data augmentation to encompass a wider range of environments, fortifying the robustness of our network. Furthermore, we aim to extend our method into downstream tasks, particularly for visual localization and mapping applications for omnidirectional images.

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

# A    3D RECONSTRUCTION

We demonstrate that our method is applicable to various omnidirectional downstream tasks, including pose estimation and 3D reconstruction. From the dense correspondences and the certainty map produced by EDM, we can estimate the essential matrix and the relative pose. Using this predicted relative pose and dense correspondences between a pair of omnidirectional images, we can construct the dense 3D reconstruction through spherical triangulation. To address spherical triangulation, we simply solve the closed-form expression (Eising, 2022),

$$\mathbf{S} \times (R(\mathbf{X} - \mathbf{C})) = \mathbf{0}, \tag{18}$$

where $\mathbf{S} = (S^x, S^y, S^z)$ is the 3D Cartesian coordinates, $R \in SO(3)$ denotes the orientation of the camera, $\mathbf{X}$ represents the target 3D point, and $\mathbf{C}$ indicates the camera position. The cross product can be expressed using a skew-symmetric matrix, leading to the following equation,

$$
\begin{aligned}
S^x \mathbf{r}^{3\mathrm{T}}(\mathbf{X} - \mathbf{C}) - S^z \mathbf{r}^{1\mathrm{T}}(\mathbf{X} - \mathbf{C}) = 0, \\
S^y \mathbf{r}^{3\mathrm{T}}(\mathbf{X} - \mathbf{C}) - S^z \mathbf{r}^{2\mathrm{T}}(\mathbf{X} - \mathbf{C}) = 0, \\
S^x \mathbf{r}^{2\mathrm{T}}(\mathbf{X} - \mathbf{C}) - S^y \mathbf{r}^{1\mathrm{T}}(\mathbf{X} - \mathbf{C}) = 0,
\end{aligned}
\tag{19}
$$

where $\mathbf{r}^{i\mathrm{T}}$ denotes the $i$th row of $R$. To determine the target 3D point $\mathbf{X}$, we can estimate the two-view geometry using the linear equation $A\mathbf{X} = \mathbf{b}$. This equation can be solved by the pseudo-inverse method, considering two omnidirectional cameras $\mathcal{M}$ and $\mathcal{N}$,

$$
A = \begin{pmatrix}
S^x_{\mathcal{M}} \mathbf{r}^{3\mathrm{T}}_{\mathcal{M}} - S^z_{\mathcal{M}} \mathbf{r}^{1\mathrm{T}}_{\mathcal{M}} \\
S^y_{\mathcal{M}} \mathbf{r}^{3\mathrm{T}}_{\mathcal{M}} - S^z_{\mathcal{M}} \mathbf{r}^{2\mathrm{T}}_{\mathcal{M}} \\
S^x_{\mathcal{N}} \mathbf{r}^{3\mathrm{T}}_{\mathcal{N}} - S^z_{\mathcal{N}} \mathbf{r}^{1\mathrm{T}}_{\mathcal{N}} \\
S^y_{\mathcal{N}} \mathbf{r}^{3\mathrm{T}}_{\mathcal{N}} - S^z_{\mathcal{N}} \mathbf{r}^{2\mathrm{T}}_{\mathcal{N}}
\end{pmatrix}, \qquad
\mathbf{b} = \begin{pmatrix}
(S^x_{\mathcal{M}} \mathbf{r}^{3\mathrm{T}}_{\mathcal{M}} - S^z_{\mathcal{M}} \mathbf{r}^{1\mathrm{T}}_{\mathcal{M}})\mathbf{C}_{\mathcal{M}} \\
(S^y_{\mathcal{M}} \mathbf{r}^{3\mathrm{T}}_{\mathcal{M}} - S^z_{\mathcal{M}} \mathbf{r}^{2\mathrm{T}}_{\mathcal{M}})\mathbf{C}_{\mathcal{M}} \\
(S^x_{\mathcal{N}} \mathbf{r}^{3\mathrm{T}}_{\mathcal{N}} - S^z_{\mathcal{N}} \mathbf{r}^{1\mathrm{T}}_{\mathcal{N}})\mathbf{C}_{\mathcal{N}} \\
(S^y_{\mathcal{N}} \mathbf{r}^{3\mathrm{T}}_{\mathcal{N}} - S^z_{\mathcal{N}} \mathbf{r}^{2\mathrm{T}}_{\mathcal{N}})\mathbf{C}_{\mathcal{N}}
\end{pmatrix}.
\tag{20}
$$

The results of 3D reconstruction are shown in Fig. 11 and Fig. 12.

# B    FURTHER QUALITATIVE RESULTS

## B.1    MATTERPORT3D

We proivde additional qualitative results for Matterport3D, as shown in Fig. 13 and Fig. 14. In Fig. 13, we present the results of RoMa (Edstedt et al., 2023b) instead of DKM, differing from the main paper.

## B.2    STANFORD2D3D

There are many occluded regions due to narrow corridors in the scenes. However, EDM, which is trained on Matterport3D, has the capability to handle these regions with certainty estimation, as shown in Fig. 15.

## B.3    EGONERF AND OMNIPHOTOS

As the environments of EgoNeRF and OmniPhotos differ significantly from the Matterport3D dataset, there is a slight performance degradation. However, comparable performance maintained with certainty estimation, as shown in Fig. 16 and 17.

756
757
758
759
760
761
762
763
764
765
766
767
768
769
770
771
772
773
774
775
776
777
778
779
780
781
782
783
784
785
786
787
788
789
790
791
792
793
794
795
796
797
798
799
800
801
802
803

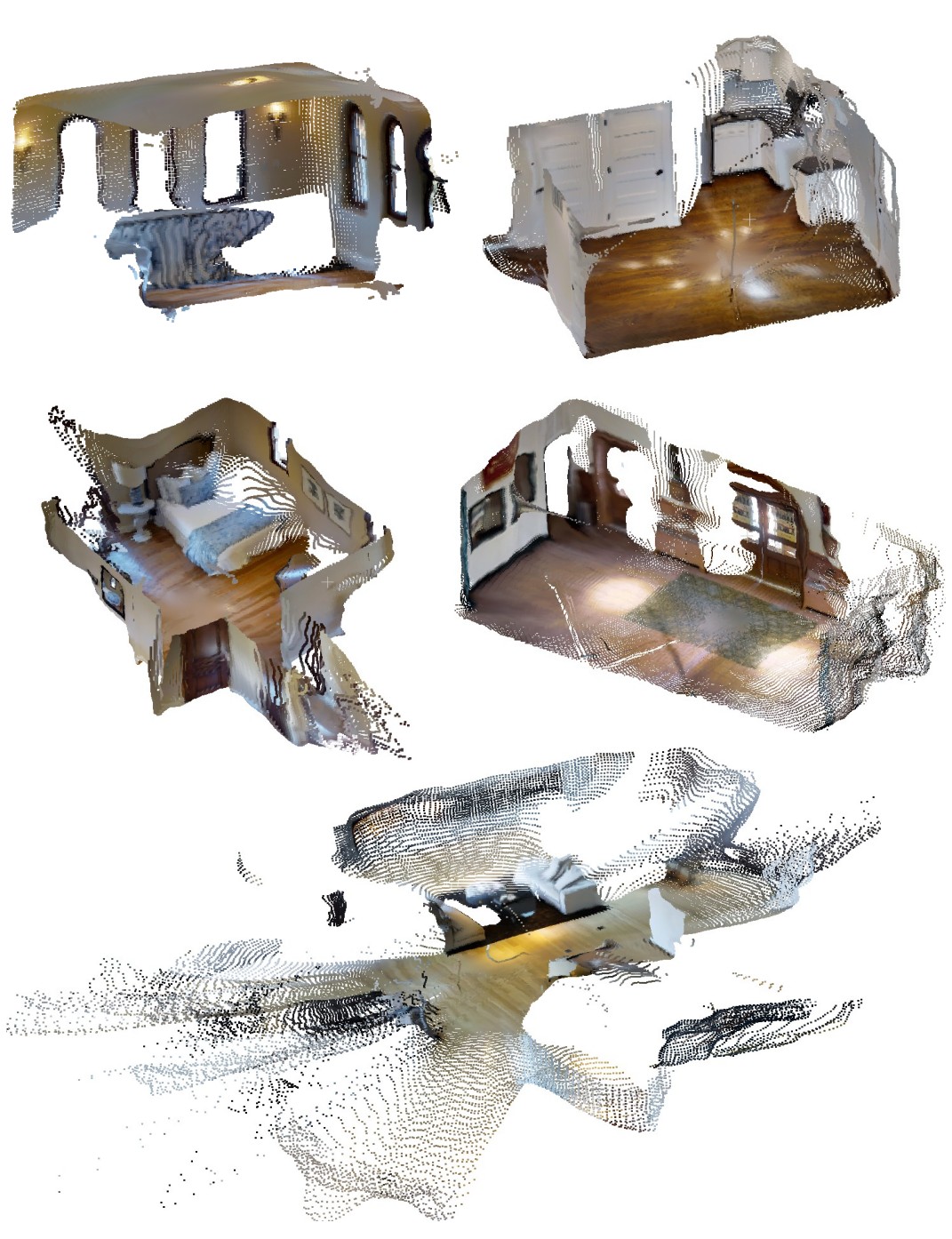

Figure 11: 3D geometry of Matterport3D using matches and certainties produced by EDM. These point clouds result from spherical triangulation with estimated poses between two omnidirectional images.

804
805
806
807
808
809

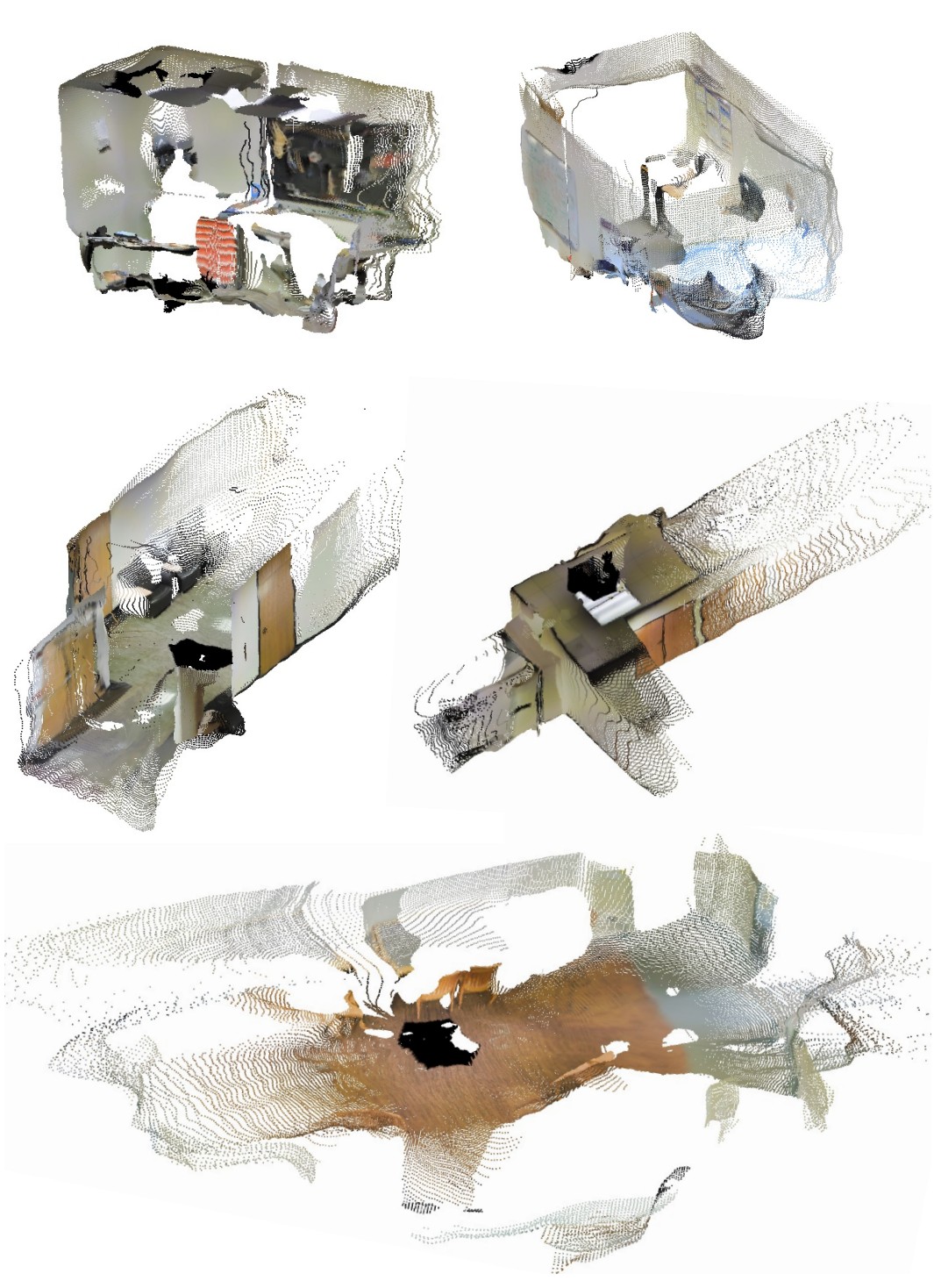

Figure 12: 3D geometry of Stanford2D3D using matches and certainties produced by EDM. These point clouds result from spherical triangulation with estimated poses between two omnidirectional images.

| Keypoint matching | RoMa | EDM (ours) | Keypoint matching | RoMa | EDM (ours) |
|---|---|---|---|---|---|

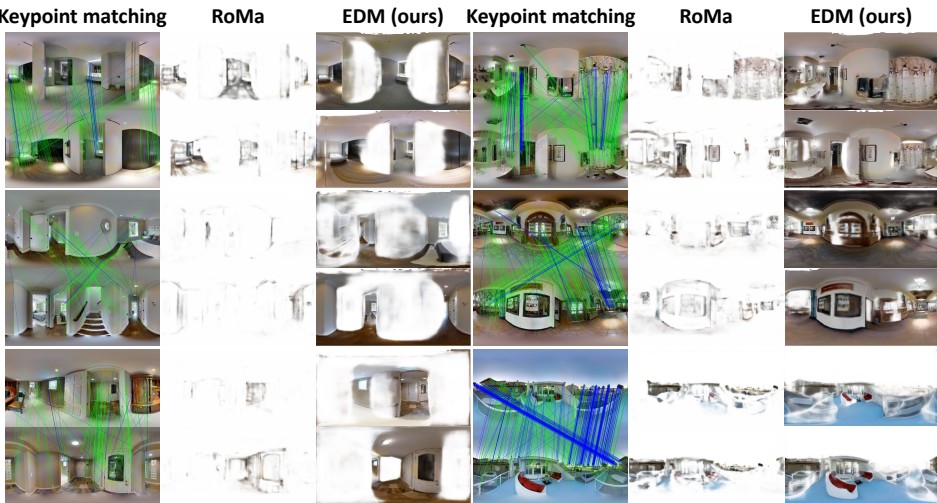

Figure 13: Qualitative results on Matterport3D. The blue lines represent the results of matching points from SPHORB (Zhao et al., 2015); the green lines correspond to SphereGlue (Gava et al., 2023). EDM demonstrates more robust performance compared to other methods.

| Image | Warp | Image | Warp |
|---|---|---|---|

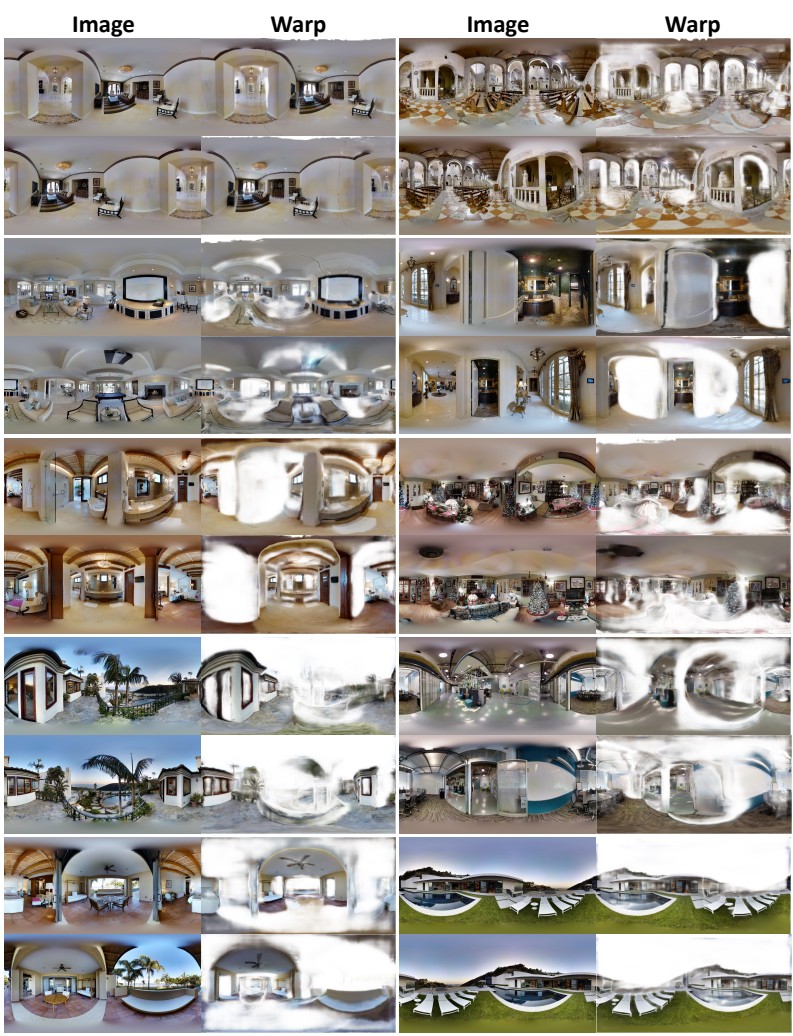

Figure 14: Qualitative results on Matterport3D.

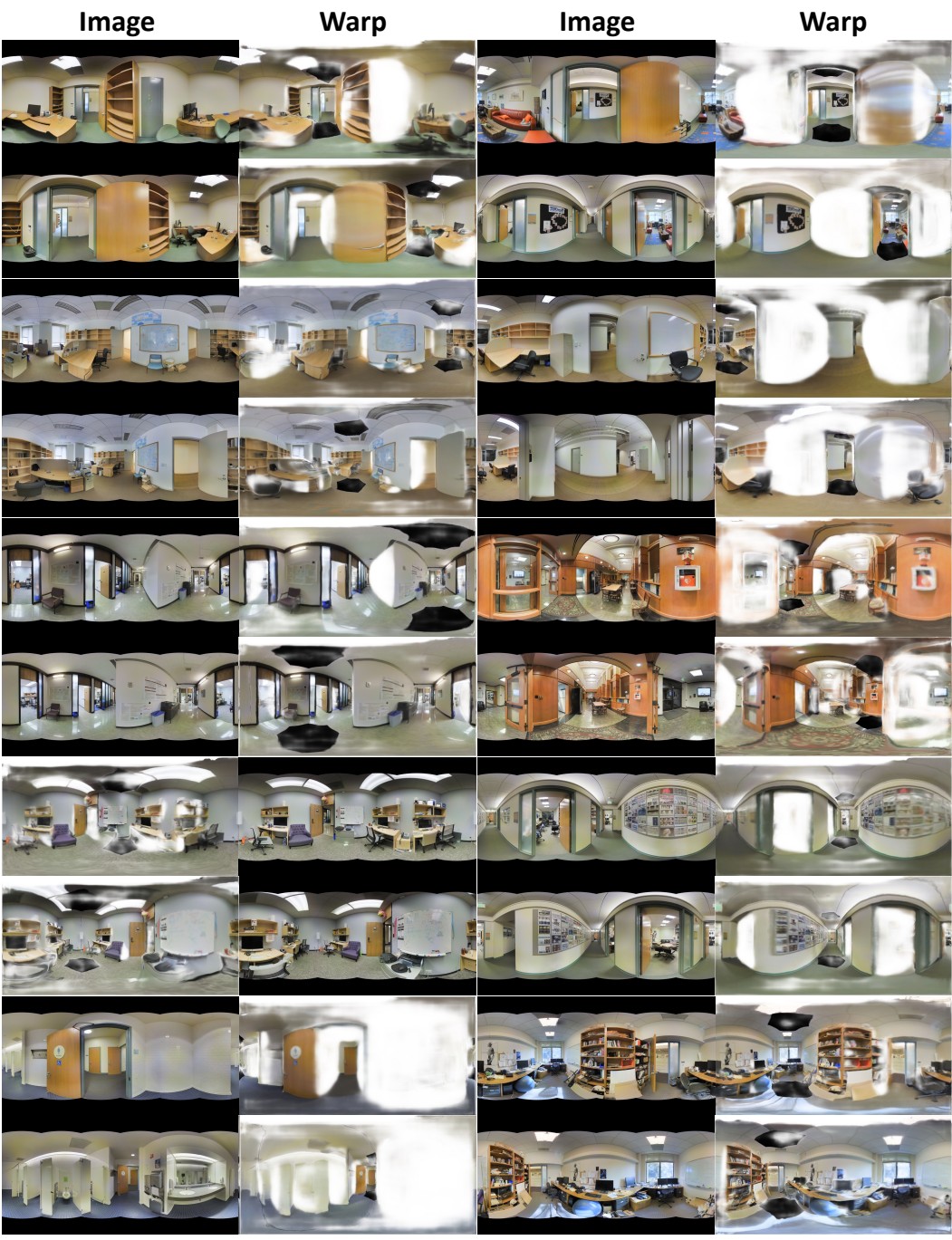

Figure 15: Qualitative results on Stanford2D3D.

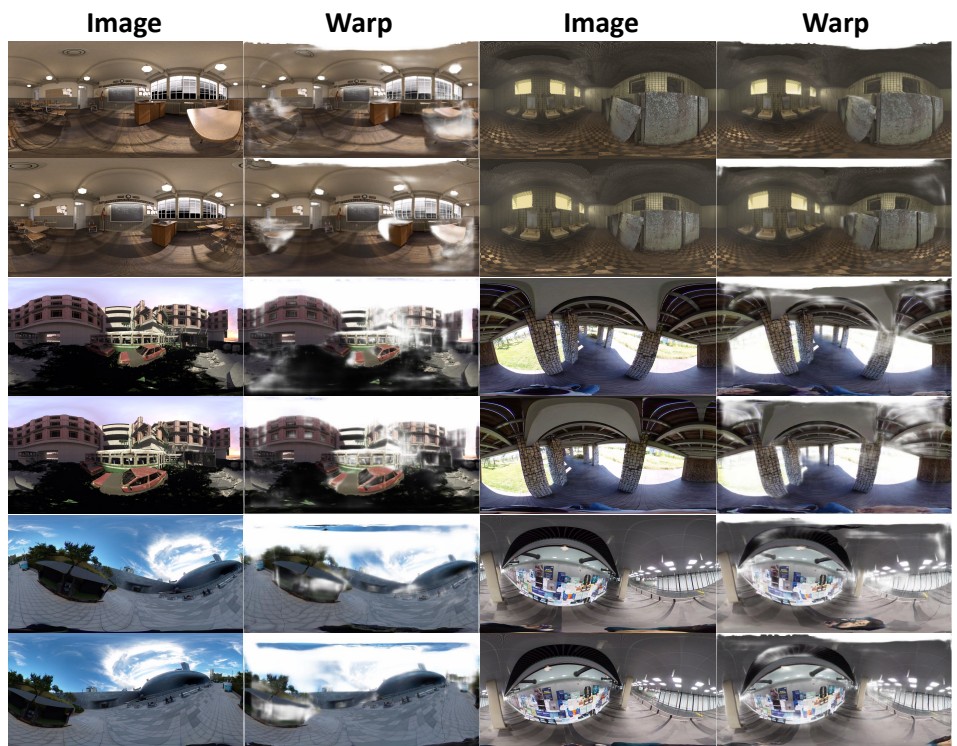

Figure 16: Qualitative results on EgoNeRF.

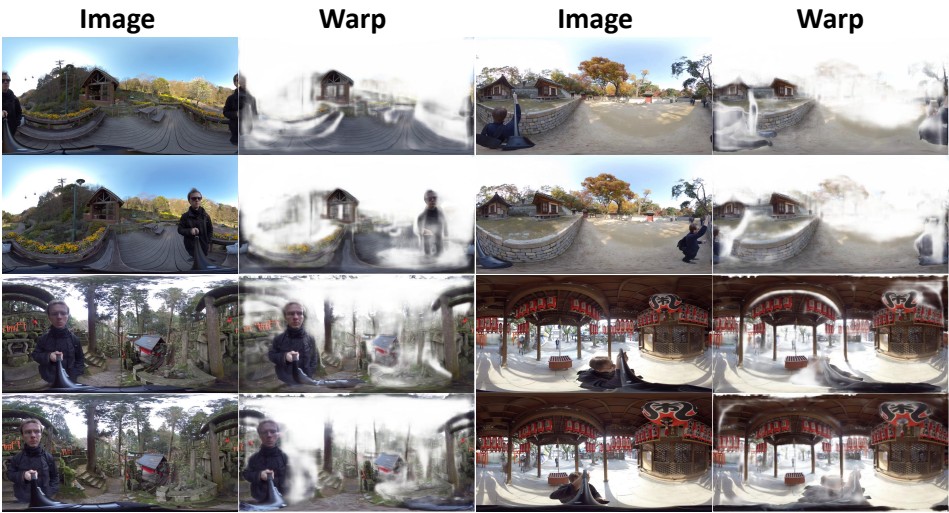

Figure 17: Qualitative results on OmniPhotos.

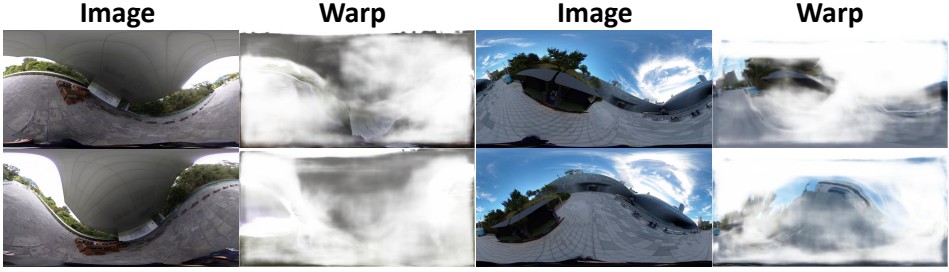

Figure 18: Failure cases.

## C    THOROUGH DISCUSSION ON LIMITATIONS AND FUTURE WORK

In this section, we provide a thorough discussion of limitations and future work associated with our study. As our work is the first to develop a dense feature matching method for omnidirectional images, we believe this discussion will advance this research direction and offer deeper insights for the 360° imaging research community.

### C.1    RUNTIME EVALUATION

EDM's runtime is almost the same as the DKM (Edstedt et al., 2023a) method because EDM includes an additional coordinate transformation between layers without requiring extra learning parameters. Both DKM and EDM take approximately 0.24 seconds per frame pair on a 3090 GPU. Comparing the runtime between sparse matching, such as SphereGlue (Gava et al., 2023) and dense matching is somewhat challenging due to differences in feature extraction and the number of matches. Sparse matching requires feature extraction before matching, and SphereGlue involves a local planar approximation to create multiple tangential images (perspective images) during feature extraction, which takes about 3.2 seconds. The inference speed for matching itself depends on the number of extracted features. In most cases, the number of features is much smaller than in dense matching, making it faster than 0.2 seconds.

### C.2    ROTATIONAL DIVERSITY IN TRAINING DATA

Our primary training dataset, Matterport3D (Chang et al., 2017), consists of indoor scenes captured with vertically fixed cameras. As a result, images with extreme rotations do not perform well in EDM, as shown in Fig. 18. We believe this problem can be mitigated by collecting more diverse training data, including images with various rotational angles, and by applying additional rotational augmentation techniques during the training process. These steps would enhance the model's ability to handle a wider range of image orientations effectively.

### C.3    ENCODER CHOICE AND DISTORTION COMPENSATION

In this paper, we use a ResNet encoder for multi-scale feature extraction. While distortion-aware approaches (Jiang et al., 2021; Wang et al., 2020; Shen et al., 2022) exist, these methods did not yield satisfactory results in our experiments and required significant computational resources. Consequently, we employed ResNet with spherical positional embeddings to compensate for distortion without adding extra trainable layers. This approach demonstrates promising results, however, feature extraction does not fully address distortion issues. In the future, we will extend our work to develop more efficient encoders capable of handling distortions.

### C.4    UTILIZATION OF FOUNDATION MODELS

In dense matching tasks for perspective images, leveraging foundation models for coarse features (Edstedt et al., 2023b) has shown better performance compared to sharing coarse-fine features using a ResNet encoder (Edstedt et al., 2023a). In this paper, our primary goal is to demonstrate the potential of a dense matching method for omnidirectional images. We believe that adopting different foundational models, as Edstedt et al. (2023b) did, could improve our framework. We plan to train foundation models such as DINOv2 (Oquab et al., 2023) or CroCo (Weinzaepfel et al., 2022) on omnidirectional images and integrate these into our approach.

