# OpenReview forum: "EDM: Equirectangular Projection-Oriented Dense Kernelized Feature Matching"
_ICLR.cc/2025/Conference — ICLR 2025 Conference Withdrawn Submission_

### Official Review · Reviewer_dXsd · 2024-10-29

**Soundness:** 3
**Presentation:** 3
**Contribution:** 3
**Rating:** 6
**Confidence:** 4

**Summary:**

The authors present an extension of the recent dense feature matcher DKM for spherical images. The main contribution is predicting the matches on the sphere instead of on a normalized cartesian grid.

**Strengths:**

- The writing of the paper was in most cases clear and easy to follow, in particular the reviewer found the figures helpful in illustrating the method.
- The proposed approach is simple and does not incur additional computational costs.
- EDM performs well on Matterport3D (which is also used for training), and Stanford2D3D (which indicates that it also generalizes). The authors have also taken care to evaluate previous sphere-based matchers on their new proposed benchmarks.

**Weaknesses:**

- EDM seems to be less robust on EgoNeRF and OmniPhotos. However, no quantitative comparison was done for those datasets. It would have been interesting to see how EDM fares against SphereGlue there.
- Some design choices were not clear to me. For example, on line 261-262 it is stated that linear refinement on the sphere is impossible, so it must be projected to equirectangular 2D space before the refinement. To me, an obvious ablation would be to compare this to a simple projection operator, i.e. $\hat{u}^{\ell} = {\rm normalize} (\hat{u}^{\ell+1} + \triangle \hat{u}^{\ell+1})$.
- Possibly minor complaint. The authors work in the DKM framework where global matching is seen as a coordinate regression problem, however it can also be seen simply in terms of dense correlation between the features (where the network would not need to "see" any embeddings). It would have been nice to see a comparison to such an approach (i.e. instead either regressing the correlation vector as in e.g. PDCNET, or using a cross-view Transformer as in LoFTR.)

**Questions:**

- In e.g. Figure 7, it can be seen that the model is quite certain about the floor. The reviewer is not certain to understand how this is possible. It's also seemingly pretty certain in the bottom example of a cupboard that does not seem to be covisible. The reviewer is wondering if the authors could explain a bit more about the confidence (under/over) of the model.
- The reviewer did not find a definition of AUC in the paper. Is it AUC of the relative pose error as in most matching works? Could be good to include in the appendix, especially as ICLR readers may be less familiar with the topic.

---

### Official Review · Reviewer_wsbZ · 2024-11-01

**Soundness:** 2
**Presentation:** 2
**Contribution:** 2
**Rating:** 3
**Confidence:** 4

**Summary:**

This paper extends the DKM method to panoramic image registration, achieving improvements across multiple datasets. However, the main contribution lies in (somewhat simple) considerations of the spherical camera model within positional encoding and matching optimization. It does not introduce new insights for the matching task itself, yielding limited technical innovation.

**Strengths:**

1. This paper combines DKM framework with omnidirectional images by considering the spherical camera model in positional encoding and correspondence optimization. It achieves dense matching omnidirectional images for the first time and achieves SoTA performances across multiple datasets.

2. The paper is well-written and designs detailed ablation experiments to verify the proposed designs.

**Weaknesses:**

1. For the reviewer, the innovation of this paper does not meet the standards of ICLR. The core algorithm is derived from DKM, introducing several optimizations for omnidirectional images (such as coordinate representation or transformation based on the spherical camera model). Although the authors demonstrated in Table 3 that these optimizations significantly improve performance over DKM in omnidirectional image matching, the work does not achieve a breakthrough in the dense matching framework, limiting its potential for broader insights and inspiration.

2. The proposed rotation augmentation strategy is designed specifically for vertically fixed cameras, suitable for indoor scenes in Matterport3D and Stanford2D3D. However, such a strategy falls short in scenarios involving extreme rotations or complex outdoor environments.

**Questions:**

My main concern lies is the technical contribution. The positional encoding from the spherical camera model lacks in-depth exploration.  Architectural or formulation designs for information exchange and matching between distorted images, as well as a more general data augmentation strategy, would be more inspiring.

---

### Official Review · Reviewer_pxyy · 2024-11-01

**Soundness:** 3
**Presentation:** 3
**Contribution:** 3
**Rating:** 6
**Confidence:** 4

**Summary:**

The paper proposes a new method for dense sphere image matching. Previous perspective image matching methods like DKM, and ROMA perform poorly when directly used for matching spherical images due to the severe image distortion. The method proposed in this paper solves this problem by introducing the spherical positional encoding into the coarse global matching of ROMA, and a refinement strategy that regresses offset on the sphere. The proposed method achieves state-of-the-art performance on the dense sphere image matching task and outperforms previous sphere matching methods and perspective matching methods by a large margin.

**Strengths:**

S1) The reviewer thinks the proposed method is reasonable and effective, which leverages the geometrical property of sphere images to improve the performance of dense matching.
S2) The paper is clearly written and well-organized. The proposed method is well-explained and easy to follow.

**Weaknesses:**

W1) Experiments. It seems that the results of baseline DKM and ROMA in Tabel 1, and 2 are obtained using their pre-trained checkpoints. However, the reviewer thinks including their results trained using the same sphere image dataset as the proposed method would be more convincing.

**Questions:**

Q1) The reviewer is curious about whether the proposed positional encoding and refinement strategy can be applied to other dense matching methods, such as ROMA.

---

### Official Review · Reviewer_hw2T · 2024-11-02

**Soundness:** 2
**Presentation:** 2
**Contribution:** 2
**Rating:** 5
**Confidence:** 3

**Summary:**

This paper proposes EDM, a learning-based dense match algorithm for omnidirectional images. Specifically, a spherical positional embeddings based 3D cartesian coordiantes and a bidirectional transformations are used to enhance the performance. The experiments on various datasets show its effectivenss.

**Strengths:**

* Utilize Gaussian Process regissin and spherical positional embedding to establish 3D correspondences between different frames.
* The refinement for geodesic flow could enhance the performance.
* The proposed method achieves better performance than baseline methods on various datasets.

**Weaknesses:**

* The novelty of the proposed method is limited since all used modules are proposed in existing methods.
* There are too many words used to describe the selected datasets in Sec. 5.1, which is not necessary.
* There are few visual results about the baseline methods and the proposed methods.
* The baselin methods do not consist of EgoNeRF in Tables 1, and 2, the most recentl method about this task.
* There is no efficient analysis about the proposed and baseline methods.

**Questions:**

* In Fig. 10, which is the results of baseline methods and the proposed methods?

---

### Note · Authors · 2024-11-15

I have read and agree with the venue's withdrawal policy on behalf of myself and my co-authors.